# Self-Calibrated Listwise Reranking with Large Language Models

## Abstract

Large language models (LLMs), with advanced linguistic capabilities, have been employed in reranking tasks through a sequence-to-sequence approach. In this paradigm, multiple passages are reranked in a listwise manner and a textual reranked permutation is generated. However, due to the limited context window of LLMs, this reranking paradigm requires a sliding window strategy to iteratively handle larger candidate sets. This not only increases computational costs but also restricts the LLM from fully capturing all the comparison information for all candidates. To address these challenges, we propose a novel self-calibrated listwise reranking method, which aims to leverage LLMs to produce global relevance scores for ranking. To achieve it, we first propose the relevance-aware listwise reranking framework, which incorporates explicit list-view relevance scores to improve reranking efficiency and enable global comparison across the entire candidate set. Second, to ensure the comparability of the computed scores, we propose self-calibrated training that uses point-view relevance assessments generated internally by the LLM itself to calibrate the list-view relevance assessments. Extensive experiments and comprehensive analysis on the BEIR benchmark and TREC Deep Learning Tracks demonstrate the effectiveness and efficiency of our proposed method.

**ACM Reference Format:**
Anonymous Author(s). 2024. Self-Calibrated Listwise Reranking with Large Language Models. In . ACM, New York, NY, USA, 10 pages. https://doi.org/10.1145/nnnnnnn.nnnnnnn

## 1 Introduction

Text reranking[1] is a fundamental task in the information retrieval (IR) area, especially in web search. It focuses on scoring and reranking a set of retrieved text candidates (*e.g.,* passages and documents) on the input query [58]. In the real world, text reranking is generally an important intermediate stage in the widely-used IR pipeline, underpinning numerous downstream tasks, such as question answering [27] and dialogue systems [52]. Concretely, this task aims to measure the semantic relevance of each text candidate with the input query, and then ranks all candidates in order of that [30, 62]. In recent years, with the exceptional problem-solving capabilities of large language models (LLMs) [59], existing work has

---

[1]Text reranking is highly related to the Search and retrieval-augmented AI Track since it is an important task in web search scenarios.

applied LLMs to the text reranking tasks [25, 46, 64]. Instead of individually computing the relevance score for all query-candidate pairs, LLM-based methods are capable of directly generating the permutation of reranked candidates in an autoregressive manner [26, 40]. Such a listwise paradigm enables efficient one-pass reranking for all candidates, and can also leverage the strong generation capability of LLMs, achieving remarkable performance.

Despite the success, limited by the input window length of LLMs, it is hard to apply listwise LLM-based rerankers into a large candidate set or long documents. Although existing work has proposed the sliding window strategy [34, 35] that splits the candidate set for multi-round ranking, the increased computational cost is also higher for real-world applications. Moreover, the sliding window strategy would cause only part of the whole candidate set to be ranked by LLM at the same time. As a result, the global ranking process will degrade into local ranking within the window, which not only restricts LLMs from fully comparing all candidates but also leads to potential risks of the influence from the initial input order. Considering these limitations, several efforts are made to optimize the sliding window strategy [33, 56] or the autoregressive generative reranking paradigm [40]. Nevertheless, as they rely on LLMs for language generation (ranked permutation), the shortcomings in efficiency and effectiveness are still hard to fully resolve.

In this paper, we aim to propose a novel method to enable LLMs to efficiently and effectively perform listwise reranking. Given the whole candidate set, our motivation is to explicitly compute the list-view relevance scores for the listwise input, instead of directly producing the textual reranking results via LLMs. In this way, the list-view relevance scores can be used for a global ranking of all candidates (in the same window or not), which breaks the shortcomings caused by the in-window local comparison. To assess the relevance, we add a projection layer into the decoder-only LLM, to map the last token representations of the candidate text into the score. For the given in-window candidates, we can obtain their relevance scores and utilize ranking objectives for training. However, the ranking objectives mainly focus on learning the comparison of all candidates, which would lead to biased scores that affect the global ranking performance, especially for the top or bottom candidates (with extreme scores of 1 or 0). To address it, we propose self-calibrated training that adjusts the list-view relevance score to better align with the self-generated point-view relevance score. The point-view relevance score is generated solely based on the query and a single candidate, which is relatively fair and provides a regularization for reducing the bias. In this way, we can make use of two views of relevance scores for supervising the training process. The list-view scores provide rich comparison information, and the point-view scores calibrate the possible bias in the list-view ones, both ensuring the global comparability of the relevance score.

To this end, we design a S̲elf-C̲alibrated L̲istwise R̲eranking method, termed SCaLR. First, we devise the relevance-aware listwise reranking framework by revising the autoregressive generation process of LLMs. Concretely, we add corresponding projection

layers for generating list-view relevance scores. To reduce the computational cost, we design a special mask mechanism to guarantee that the list-view relevance scores can be computed by a one-pass encoding process [57]. Based on this, we introduce the parallel context encoding into the decoder-only LLM architecture, enabling independent encoding of candidates to generate the point-view relevance scores. Second, we propose the self-calibration training strategy that aligns the list-view relevance score with the point-view relevance score. Specifically, we employ the multi-task learning framework to learn both list-view and point-view relevance scores, and propose an adaptive optimization strategy to consider the reliability of the point-view scores during calibration.

Our main contributions are summarized as follows:

- We propose a novel listwise reranking framework SCaLR based on explicit list-view relevance for ranking, which enhances the model efficiency while addressing the limitation of window-based local ranking strategies.
- We employ parallel context encoding for accelerating candidate modeling and utilize self-generated point-view relevance to calibrate the list-view relevance, ensuring global comparability for evaluating on large candidate set.
- Extensive experiments and analyses on the BEIR and TREC benchmarks demonstrate the superiority of the proposed approach from in-domain and out-of-domain evaluation over state-of-the-art methods.

## 2 Preliminaries

Before diving into the details of the proposed method, we first formulate the task of text reranking, followed by a formal definition of the listwise reranking task in the era of large language models (LLMs).

Let $q$ denote a query in natural language form and $C = \{p_i \mid i \in \{1, 2, \ldots, |C|\}, p_i \in \mathcal{P}\}$ denote a set of candidate texts relevant to $q$. $C$ is retrieved from a large-scale candidate corpus $\mathcal{P}$ by a retrieval model (usually for hundreds or thousands of candidates). The task of text reranking is to refine the order of candidates in $C$ by leveraging more granular relevance modeling, ultimately producing a ranking order that more accurately reflects the real relevance between the candidates and the query, which can be formalized as:

$$C' = \{p_{i_1}, p_{i_2}, \ldots, p_{i_{|\mathcal{L}|}}\} = \text{Rerank}(C). \quad (1)$$

In traditional reranking tasks, the reranker independently evaluates the relevance between the query $q$ and each candidate $p_i$, using this relevance assessment to determine a new ranking order. This approach is referred to as the *pointwise reranking* method.

Given that LLMs possess advanced linguistic capabilities that can process multiple candidates simultaneously, LLM-based rerankers typically rerank a subset $C_{sub}$ of the candidate set $C$ in a single inference with a well-designed instruction $I$:

$$C'_{sub} = \text{LLM}(I, C_{sub}), \quad (2)$$

where the output of the LLM is typically a textual response that indicates the reranked order of the candidate ids. Subsequently, the reranking results on the complete candidate set $C$ can be obtained based on multiple iterations with subset reranking through tailored strategies (*e.g.*, sliding window). Such a reranking method is referred

to as the *listwise reranking*, which leverages contextual information more effectively, thereby improving overall ranking performance.

Although the listwise reranker has demonstrated commendable performance in reranking tasks, its inference efficiency remains a significant bottleneck. This limitation primarily arises from its dependence on strategies such as sliding windows for iterative reranking across the entire set of candidates, coupled with its autoregressive generation framework, which substantially increases the model's redundant computations. Furthermore, the sliding window strategy ensures ranking accuracy only within the window size, failing to guarantee global optimality over extended ranges, thus restricting its applicability in scenarios requiring longer reranking sequences. In contrast to previous approaches, our method improves the architecture of the listwise reranker and introduces explicit relevance assessments for global comparability. This not only improves computational efficiency but also achieves global optimality across the entire candidate set $C$.

## 3 Methodology

In this section, we present SCaLR, our method for listwise reranking, which introduces two key technical contributions: (1) the relevance-aware listwise reranking framework for incorporating explicit relevance with high efficiency, and (2) the self-calibration approach for adaptively calibrating the list-view relevance by internal signals of the model.

### 3.1 Relevance-Aware Listwise Reranking Framework

The prevailing listwise reranking paradigm, which generates ranking permutations in textual form, faces inherent limitations in terms of both efficiency and local optima. To overcome these challenges, we propose an enhancement to this framework by incorporating explicit list-view and point-view relevance in one LLM.

*3.1.1 List-view Relevance Incorporation.* Based on cross-attention modeling within LLM, we concatenate the candidate identifiers of all candidates to the end of the input sequence. A linear projection layer is then applied over the output layer, ultimately producing a real-valued score that quantifies the relevance of each candidate to the query. The green part on the left side of Figure 1 illustrates this concept. The formal definition is as follows:

$$ls_i = \text{Linear}_{d \to 1}\left(\text{LLM}(\text{input})[idx_i^{\text{id}}]\right), \quad (3)$$

where $idx_i^{\text{id}}$ denotes the position of the concatenated $i$-th candidate identifier within the input sequence. For a listwise input comprising $M$ candidates, the LLM generates a relevance list $\{ls_i\}_{i=1}^{M}$ corresponding to the input candidates.

Since this relevance is derived from listwise input, we define it as *list-view* relevance. In our listwise reranking paradigm, all candidate identifiers are concatenated at the end of the input sequence. To ensure that the generation of each list-view relevance score remains independent of previous candidate identifiers, we introduce an attention mask matrix at the position of the candidate identifiers to block information exchange between them.

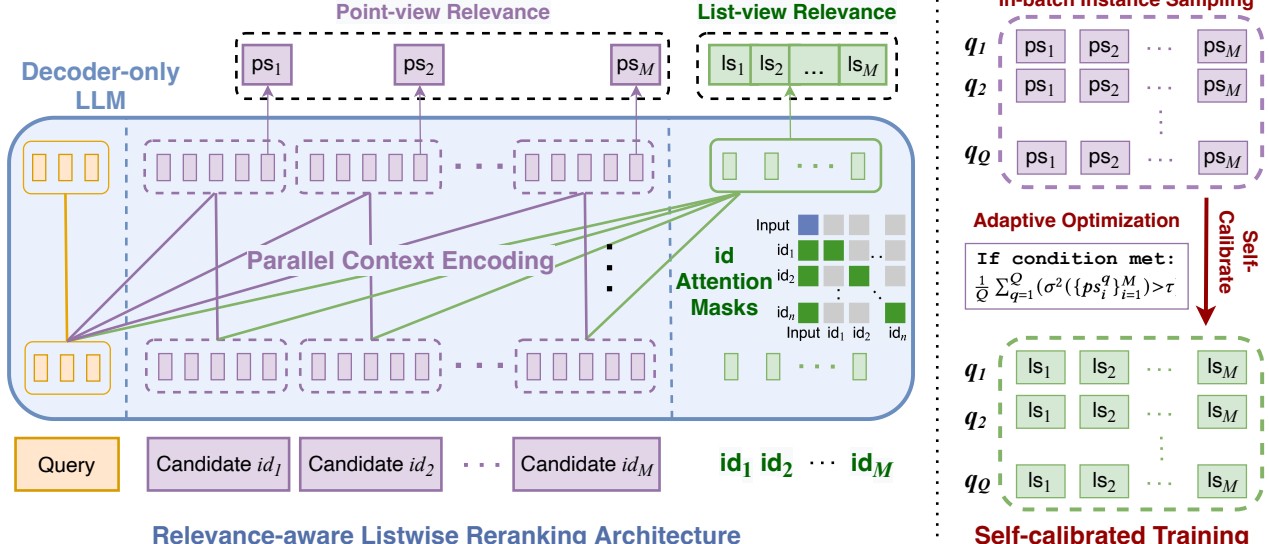

**Figure 1: The proposed SCaLR method for listwise reranking. The relevance-aware listwise reranking architecture generates list-view relevance scores to capture information from multiple candidates and uses parallel context encoding to generate point-view relevance scores for each candidate. During self-calibrated training, the list-view relevance is adaptively calibrated by the point-view relevance with the in-batch sampling strategy.**

*3.1.2 Point-view Relevance with Parallel Context Encoding.* List-view relevance offers a comprehensive relevance evaluation of cross-candidate information in the current listwise input. Furthermore, we introduce parallel context encoding [55] into the decoder-only LLM to enable relevance capture merely based on a single candidate in listwise inputs, since such a pointwise relevance naturally satisfies global optimal. The core idea behind this approach is to assign identical positional encodings to all input candidates, ensuring that each candidate is modeled independently, without interference from other candidates. The left part in the purple of Figure 1 provides an illustration. We append a special token <DOC_END> at the end of each candidate and incorporate a linear mapping layer at this position to generate a relevance score:

$$ps_i = \text{Linear}_{d \rightarrow 1}\Big(\text{LLM}(\text{input})[idx_i^{\text{st}}]\Big), \qquad (4)$$

where $idx_i^{\text{st}}$ denotes the position of the special token of $i$-th candidate in the input sequence. Since each relevance score is assessed based on a single candidate, we refer to it as *point-view* relevance.

The incorporation of parallel context encoding ensures that independent point-view relevance scores for every query-candidate pair can be extracted within the LLM, without being influenced by the presence of other candidates. This establishes a crucial foundation for the subsequent calibration process.

## 3.2 Self-Calibrated Training

*3.2.1 Learning for List-view and Point-view Relevance.* By introducing explicit list-view relevance scores, the model's optimization shifts from a text generation task to a similarity-based ranking task. To ensure fairness in subsequent comparisons, we adhere to established research and utilize the textual ranking permutation

generated by RankGPT [46] as the training data source. Consequently, we employ the RankNet [2] loss to optimize the model in a pairwise manner:

$$\mathcal{L}_{\text{List}} = \sum_{i=1}^{M}\sum_{j=1}^{M} \mathbf{1}_{r_i < r_j} \log(1 + \exp(ls_i - ls_j)), \qquad (5)$$

where $r_i$ and $r_j$ represent the ranks of $i$-th and $j$-th candidates in the permutation, respectively. Since RankNet is a pairwise loss function, we decompose the entire permutation into all possible pairwise combinations, resulting in $M(M-1)/2$ candidate pairs for training. We similarly employ RankNet loss to optimize the task of generating point-view relevance scores:

$$\mathcal{L}_{\text{Point}} = \sum_{i=1}^{M}\sum_{j=1}^{M} \mathbf{1}_{r_i < r_j} \log(1 + \exp(ps_i - ps_j)). \qquad (6)$$

*3.2.2 Self-Calibrating List-view Relevance by Point-view Relevance.* Although our relevance-aware listwise reranking paradigm theoretically ensures global comparability across different sublists, empirical attempts reveal that after training with the ranking loss, the list-view relevance scores assigned to each sublist's candidates still exhibit tendencies toward local optima. While point-view relevance demonstrates strong global optimality, it neglects the information from other candidates present in the listwise input. To overcome this challenge, We propose to utilize the global optimal characteristics of point-view relevance scores to calibrate list-view relevance scores. Since point-view relevance scores are generated based on internal model parameters, this process is referred to as self-calibration.

**Self-Calibration Loss.** In our study, the objective of calibration is to ensure that the list-view relevance more accurately reflects the true relevance scale between a query and its corresponding candidate, rather than simply maintaining the relative ranking of the current candidate list. Given that point-view relevance is determined without interference from other candidate information, it provides an ideal assessment for global comparison without the need for external supervisory signals. We substitute the permutation labels used in RankNet with the point-view relevance scores, thereby calibrating the score of list-view relevance:

$$\mathcal{L}_{\text{Cal}} = \sum_{i=1}^{M} \sum_{j=1}^{M} \mathbf{1}_{(ps_i > ps_j)} \log(1 + \exp(ls_i - ls_j)). \quad (7)$$

**Adaptive Optimization using In-batch Instances.** As reranking loss focuses solely on optimizing the relevance scores of different candidates for the same query, we first propose to adopt the in-batch sampling strategy [19, 38] widely utilized in dense retrieval into the reranking task. In contrast to in-batch sampling for dense retrieval, which relies on in-batch candidates for optimization, we propose incorporating additional query-candidate pairs from the current batch into the optimization process to enlarge the instance number. This can be regarded as a global cross-query sampling strategy to achieve better calibration performance. Formally, we define the in-batch self-calibration loss as follows:

$$\mathcal{L}_{\text{Cal-IB}} = \sum_{i=1}^{M*Q} \sum_{j=1}^{M*Q} \mathbf{1}_{(ps_i > ps_j)} \log(1 + \exp(ls_i - ls_j)), \quad (8)$$

where $Q$ denotes the query number in the mini-batch. This strategy extends the calibration process from in-query to cross-query setting.

Furthermore, the point-view relevance scores are derived from the LLM's internal parameters. During the initial stage of training, point-view relevance scores may exhibit limited discrimination across different query-candidate pairs, resulting in suboptimal calibration. To address this challenge, we further introduce an adaptive optimization mechanism. Specifically, we first calculate the variance of the point-view relevance scores for the candidates of each query in the current batch, followed by calculating the average variance across all queries:

$$\overline{\text{Var}} = \frac{1}{Q} \sum_{q=1}^{Q} \sigma^2(\{ps_k^q\}_{k=1}^{M}), \quad (9)$$

where $\sigma^2(\cdot)$ denotes the operation of calculating variance. We set a threshold $\tau$ to assess whether the current point-view relevance evaluation is sufficiently reliable for the calibration task. The integral calibration loss with in-batch sampling and adaptive optimization is defined as follows:

$$\mathcal{L}_{\text{Cal-AdaIB}} = \sum_{i=1}^{M*Q} \sum_{j=1}^{M*Q} \mathbf{1}_{((\overline{\text{Var}} > \tau) \wedge (ps_i > ps_j))}$$
$$\log(1 + \exp(ls_i - ls_j)). \quad (10)$$

As a result, self-calibration is applied only after point-view relevance reaches a state deemed most favorable for adjustment, thereby avoiding potential misalignment during the initial stages of training.

*3.2.3 Final Loss.* Building on the point-view and list-view relevance optimization methods outlined in Section 3.2.1, alongside the self-calibration loss with adaptive optimization and in-batch sampling strategy presented in Section 3.2.2, we combine the list-view, point-view, and self-calibrated loss as the final optimization objective of SCaLR:

$$\mathcal{L}_{\text{Final}} = \mathcal{L}_{\text{List}} + \mathcal{L}_{\text{Point}} + \mathcal{L}_{\text{Cal-AdaIB}}. \quad (11)$$

## 3.3 Overview and Discussion

In this section, we provide an overview of SCaLR and discuss its advantages.

During inference, we only need the list-view relevance scores for reranking all candidates. Given that the listwise input comprises $M$ candidates, we split the entire candidate set $C$ into $|C|/M$ listwise inputs and compute the list-view relevance scores for the candidate texts within each input. Since our framework modifies the autoregressive listwise reranking paradigm, the generation of each list-view relevance score depends solely on the original input. It enables us to obtain relevance scores for the entire candidate set with lower cost than the sliding window strategy in existing LLM-based ranking methods. Furthermore, drawing inspiration from Quiet-STaR [57], we parallelize the generation of all relevance scores to accelerate inference and optimize memory usage, further improving efficiency and saving cost.

Our self-calibrated listwise reranking method owns two major advantages:

(1) The proposed relevance-aware listwise reranking framework introduces explicit relevance scores, enabling the model to consider both list-view and point-view relevances for the given candidates during training. The list-view relevance scores effectively capture the listwise knowledge, and the point-view relevance scores characterize the query-candidate semantic relevance. Both support using parallel training and inference techniques for acceleration.

(2) We self-calibrate the list-view relevance scores using point-view relevance scores without the need of external signals. It allows the list-view relevance to maintain the multi-candidate view while ensuring the global comparability characteristics of point-view relevance.

## 4 Experiments and Analysis

In this section, we first outline the experimental setup, followed by the presentation of the main results, ablation study, and in-depth analysis.

### 4.1 Experimental Settings

*4.1.1 Datasets.* To eliminate the impact of training data variability on model performance, we wholly adhere to current state-of-the-art listwise reranking approaches [35, 40], employing listwise comparison data generated by RankGPT-3.5 and RankGPT-4 [46] as the distillation training data. This dataset includes 100K queries annotated by RankGPT-3.5 and sampled 5K queries annotated by RankGPT-4 from MS MARCO [29], each paired with 20 candidates retrieved by the retriever, serving as input. Due to the presence of annotation noise identified in the data from RankGPT-3.5, we exclude approximately 13% of this dataset. In contrast to prior studies,

| | Zero-shot Rerankers | | | | | Fine-tuned Rerankers | | | | | | | | |
|---|---|---|---|---|---|---|---|---|---|---|---|---|---|---|
| | BM25 | RG-S | Set-wise | GPT-3.5 | GPT4 | mono-T5 | LiT5 | List-T5 | PE-Rank | Rank-LLaMA | Rank-Vicuna | Rank-Zephyr | FIRST | SCaLR (ours) |
| Model size | - | ? | 11B | ? | ? | 110M | 770M | 3B | 7B | 7B | 7B | 7B | 7B | 7B |
| Retriever | - | BM25 | BM25 | BM25 | BM25 | BM25 | BM25 | BM25 | BM25 | RepL. | Cont. | Cont. | Cont. | Cont. |
| Top-$k$ for rank | - | 100 | 100 | 100 | 100 | 100 | 100 | 100 | 100 | 200 | 100 | 100 | 100 | 100 |
| Training data | - | - | - | - | - | $M_l$ | $M_d$ | $M_l$ | $M_d$ | $M_l$ | $M_d$ | $M_d$ | $M_d$ | $M_d$ |
| Inf. strategy | PW | PW | SW | LW | LW | PW | LW | LW | LW | PW | LW | LW | LW | LW |
| Climate-FEVER | 16.5 | - | - | - | - | 23.1 | 19.8 | 24.8 | - | 28.0 | 28.2 | 25.6 | 26.7 | 23.0 |
| DBPedia | 31.8 | 41.9 | 42.4 | 44.5 | 47.1 | 42.8 | 43.5 | 46.2 | 40.1 | 48.3 | 50.0 | 50.0 | 50.9 | 50.7 |
| FEVER | 65.1 | - | - | - | - | 78.4 | 73.9 | 82.0 | - | 83.9 | 81.0 | 80.1 | 81.7 | 86.0 |
| FiQA | 23.6 | - | - | - | - | 39.2 | 41.6 | 45.1 | - | 46.5 | 35.9 | 42.2 | 42.2 | 47.6 |
| HotpotQA | 63.3 | - | - | - | - | 71.2 | 70.9 | 75.6 | - | 75.3 | 73.5 | 71.6 | 74.2 | 75.2 |
| NFCorpus | 32.2 | 39.0 | 34.6 | 35.6 | 38.5 | 35.7 | 35.4 | 37.7 | 36.4 | 30.3 | 33.1 | 37.7 | 37.4 | 38.9 |
| NQ | 30.6 | - | - | - | - | 52.1 | 55.3 | 56.2 | - | 66.3 | 58.6 | 65.6 | 66.4 | 68.1 |
| SCIDOCS | 14.9 | - | - | - | - | 16.7 | 18.1 | 19.5 | - | 17.8 | 18.4 | 20.5 | 20.4 | 21.9 |
| SciFact | 67.9 | 75.2 | 75.4 | 70.4 | 75.0 | 73.1 | 74.1 | 77.0 | 69.4 | 73.2 | 70.5 | 76.7 | 74.6 | 76.7 |
| TREC-COVID | 59.5 | 80.5 | 76.8 | 76.7 | 85.5 | 78.3 | 80.3 | 84.7 | 77.7 | 85.2 | 71.3 | 78.4 | 78.8 | 79.4 |
| Average | 43.7 | - | - | - | - | 51.1 | 51.3 | 54.9 | - | 55.5 | 52.1 | 54.8 | 55.3 | $56.8^{\dagger}$ |

**Table 1: NDCG@10 Results on BEIR. $M_l$ denotes training on MS MARCO labeled data, while $M_d$ refers to training on the distilled data. RepL. and Cont. denote RepLLaMA and Contriever, respectively. LW and PW denote listwise and pointwise reranker, respectively. We provide the experimental details of each baseline that correspond to the reported results. The symbol "$\dagger$" denotes that the performance improvement is statistically significant with p < 0.05 compared against all the baselines.**

we do not artificially expand RankGPT-4 data by sublist sampling to increase the dataset size.

For evaluation, we used the BEIR benchmark [49] to perform out-of-domain evaluations. The BEIR benchmark comprises datasets across a diverse set of datasets spanning multiple domains, designed for information retrieval tasks. We followed the evaluation settings from the previous work [40], selecting the 10 datasets for comparison consistency including Climate FEVER, DBPedia, FEVER, FiQA, HotpotQA, NFCorpus, Natural Questions, Scidocs, SciFact, and Trec-COVID. Additionally, we evaluate SCaLR utilizing the TREC Deep Learning (DL) Track [4–7] test collections from 2019, 2020, 2021, and 2022. Notably, DL 2019 and 2020 utilize the passage corpus of MS MARCO v1, while DL 2021 and 2022 rely on MS MARCO v2 passage corpus.

*4.1.2 Evaluation Metrics.* Following previous studies [35, 40], the evaluation metric utilized in our experiment is Normalized Discounted cumulative gain (NDCG). NDCG is a widely recognized measure for assessing the quality of ranked results in information retrieval tasks, by comparing the weighted relevance of all results to an ideal order. Specifically, we use NDCG@10 for evaluation in our experiment.

*4.1.3 Baselines.* Here, we compare our proposed approach against numerous competitive reranking baselines, including a sparse reranker BM25 [20], zero-shot rerankers, and fine-tuned rerankers.

Zero-shot rerankers in our study denote rerankers that directly rely on pure LLM (*e.g.,* LLaMA [50]) for evaluation and are not fine-tuned on any reranking tasks. Concretely, we adopt LRL [26], Setwise [64], RG-S (0, 4) [61], RankGPT-3.5 [46], and RankGPT-4 [46] for evaluation.

Fine-tuned reranker baselines including pre-trained language models like and T5 [39], also with LLMs-based rerankers. We introduce a diverse set of baselines to comprehensively demonstrate the superiority of our method, including monoT5 [31], LiT5 [47], ListT5 [56], RankLLaMA [25], RankVicuna [34], RankZephyr [35], PE-Rank [23], and FIRST [40]. We use PE-Rank$_{jina}$ for BEIR evaluation and PE-Rank$_{BGE}$ for TREC DL evaluation, we adopt the LiT5-Distill version for evaluation.

For consistency in the evaluation of benchmarks, we replicate some of the reported performance metrics in the baseline papers for comparison. Note that some baselines are not evaluated on specific baselines, resulting in the absence of some performance results. Given that different approaches may employ distinct first-stage retrievers to obtain candidate results, we annotate the retriever used by each approach in the table accordingly.

*4.1.4 Implement Details.* For a fair comparison, following recent competitive methods [35, 40], we adopt Zephyr$_\beta$ [51] as the backbone model for training, and use Contriver [16] to retrieve top 100 candidates for reranking on BEIR and BM25 to retrieve top 100 candidates for reranking on TREC DL. We fine-tune the model for 3 epochs with a batch size of 8 and a learning rate of 1e-5 using bfloat16 precision. Our training is implemented on eight NVIDIA A100 80G GPUs for 18 hours. We set the threshold $\tau$ of 10 to adaptively introduce the calibration loss considering the average variation.

## 4.2 Main Results

In this section, we present the evaluation results of SCaLR on both the BEIR and TREC DL Tracks.

| | Model Size | Source Retriever | top-$k$ | Training Data | DL19 nDCG@10 | DL20 nDCG@10 | DL21 nDCG@10 | DL22 nDCG@10 |
|---|---|---|---|---|---|---|---|---|
| *Zero-shot Rerankers* | | | | | | | | |
| BM25 [20] | - | - | - | - | 50.6 | 48.0 | 44.6 | 26.9 |
| LRL [26] | ? | BM25 | 100 | - | 65.8 | 62.2 | 60.0 | - |
| Setwise [64] | 11B | BM25 | 100 | - | 71.1 | 68.6 | - | - |
| GPT-3.5 [46] | ? | BM25 | 100 | - | 65.8 | 62.9 | 60.1 | 41.8 |
| GPT-4 [46] | ? | BM25 | 100 | - | 73.6 | 70.6 | 70.7 | 50.8 |
| *Fine-tuned Rerankers* | | | | | | | | |
| monoT5 [31] | 3B | BM25 | 100 | $M_l$ | 71.5 | 68.9 | - | - |
| ListT5 [56] | 3B | BM25 | 100 | $M_l$ | 71.8 | 69.1 | - | - |
| PE-Rank [23] | 7B | BGE | 100 | $M_d$ | 72.9 | 67.8 | - | - |
| RankVicuna [34] | 7B | BM25 | 100 | $M_d$ | 68.5 | 69.0 | 66.1 | 43.9 |
| RankZephyr [35] | 7B | BM25 | 100 | $M_d$ | 73.7 | 70.7 | 69.6 | 51.4 |
| SCaLR (Ours) | 7B | BM25 | 100 | $M_d$ | $74.6^{\dagger}$ | $71.0^{\dagger}$ | $71.8^{\dagger}$ | $52.1^{\dagger}$ |

Table 2: Results on TREC DL Tracks. $M_l$ denotes training on MS MARCO labeled data, while $M_d$ refers to training on the distilled data. We provide the experimental details of each baseline that correspond to the reported results. The symbol "$\dagger$" denotes that the performance improvement is statistically significant with $p < 0.05$ compared against all the baselines.

4.2.1 *Results on BEIR.* The results of different reranking methods evaluated on BEIR are shown in Table 1. It can be observed that:

(1) Among all methods, the proposed SCaLR method outperforms other baselines in average evaluation across the BEIR benchmark, demonstrating its superior out-of-domain performance. The key distinction of our method lies in the incorporation of explicit list-view relevance for performing listwise reranking. SCaLR combines the efficiency advantage of pointwise methods with the enhanced contextual understanding advantage of listwise methods. Through self-calibrated training, the list-view relevance scores are refined to be more accurate for achieving global optimum across the entire candidate set.

(2) Although the BEIR benchmark is designed for out-of-domain evaluation, our analysis indicates that zero-shot rerankers overall underperform compared to fine-tuned rerankers, even when leveraging LLMs with substantial parameter scales. We hypothesize that this performance gap stems from the absence of explicit reranking task training during the pretraining stages of LLMs without supervised fine-tuning. Nevertheless, RankGPT-4 remains a robust and competitive baseline, owing to its outstanding problem-solving capabilities.

(3) In comparison to the pointwise reranking approaches, listwise reranking methods typically superior superior performance. This can be attributed to the capabilities of LLMs to process multiple documents simultaneously, enabling more effective handling of listwise inputs and consequently yielding improved results. It is also noteworthy that RankLLaMA exhibits exceptional performance. We note that its results are derived from reranking the top 200 outputs of RepLLaMA, which may offer a certain advantage over reranking other retrieval results.

4.2.2 *Results on TREC DL Tracks.* For a fair comparison, we employ BM25 as the retriever to provide search results across multiple baselines. The results of different reranking methods evaluated on TREC DL Tracks are presented in Table 2. It can be observed that the proposed SCaLR method consistently outperforms all baselines

| Variants | BEIR Average |
|---|---|
| SCaLR | **56.8** |
| w/o adaptive optimization | 55.2 |
| w/o in-batch instances | 52.9 |
| w/o self-calibration | 52.1 |
| w/ point-view relevance | 55.3 |

Table 3: NDCG@10 results on variants of SCaLR, we report the average results across datasets in BEIR.

across all datasets. Since the datasets in the TREC DL Tracks are derived from MS MARCO v1 and MS MARCO v2, these experiments represent in-domain evaluations. The superior performance of SCaLR highlights its remarkable in-domain capabilities, which can be attributed to our calibrated, globally comparable list-view relevance score enabled by its novel listwise reranking architecture. In addition, we observe that RankGPT-4 shows compatible performance in the reranking task, even without supervised fine-tuning, leveraging its robust instruct-following capabilities to outperform many fine-tuned rerankers.

## 4.3 Effect of Relevance Calibration

In this section, we conduct an ablation study to comprehensively examine the effectiveness of key strategies in SCaLR. We report the average results on BEIR. Here, we consider three variants based on SCaLR for comparison: (a) *w/o adaptive optimization* always introduces the self-calibration loss; (b) *w/o in-batch instances* removes the optimization over other query-candidate instances within the same mini-batch; (c) *w/o self-calibration* eliminates the calibration of list-view relevance scores; (d) *w/ point-view* employs point-view relevance scores for reranking, replacing the list-view scores.

Table 3 presents the results for variants of SCaLR, from which we can observe the following findings: (a) The performance drops in

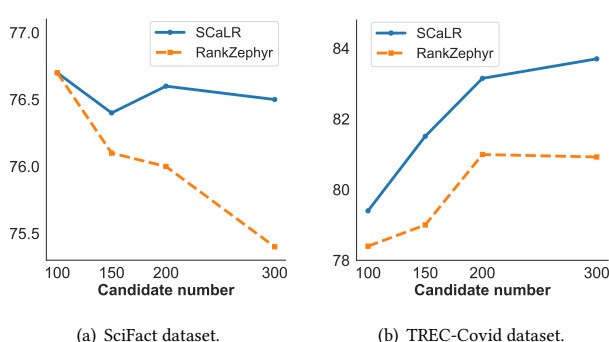

(a) SciFact dataset.  (b) TREC-Covid dataset.

**Figure 2: NDCG@10 results for extending candidate number on two BEIR datasets.**

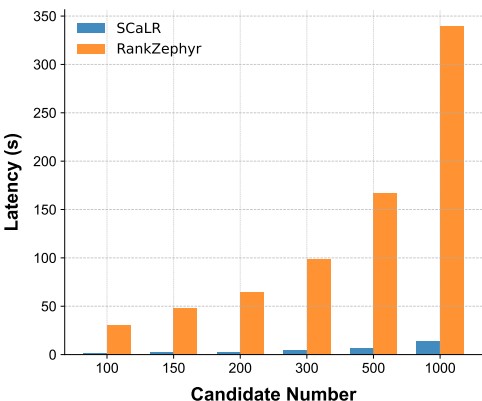

**Figure 3: The latency for reranking one query with varying numbers of candidates evaluated on TREC-Covid.**

_w/o adaptive optimization_, demonstrating that adaptively incorporating self-calibration loss during training helps list-view relevance be calibrated in a more proper opportunity. (b) The performance drops in _w/o in-batch instances_, demonstrating the importance of the importance of introducing cross-query supervision signals for relevance calibration. (c) The performance significantly drops in _w/o self-calibration_, highlighting the necessity of calibrating the list-view relevance and the effectiveness of our proposed calibration method. (d) The performance drops in _w/ point-view_, demonstrating the effectiveness of reranking with multiple candidate information.

### 4.4 Reranking on Expansive Candidate Sets

Compared to existing LLM-based listwise reranking methods, a key advantage of our proposed approach lies in the utilization of calibrated relevance scores, which replace the permutation generation used solely for local ranking. We hypothesize that this enables the method to exhibit greater robustness and efficiency across larger candidate sets. In this section, we design experiments to validate this hypothesis from the perspectives of both performance and efficiency.

_4.4.1 Evaluation on Performance._ We first investigate the scalability of our method in handling larger candidate sets by examining its reranking performance. We select RankZephyr [35], a well-established listwise reranking method based on open-source LLMs, as a representative LLM-based listwise reranker for comparison. Starting with a candidate set size of 100, we progressively increased the number of candidates to observe how different methods performed under varying conditions.

As shown in Figure 2, the evaluation results on the SciFact and TREC-Covid datasets reveal distinct patterns. On the SciFact dataset, the performance gap between SCaLR and RankZephyr widens as the number of candidates increases. On the TREC-Covid dataset, RankZephyr's overall performance declines with a larger candidate set, whereas SCaLR's performance remains relatively stable. This observation demonstrates that SCaLR exhibits superior robustness on reranking performance when reranking a larger number of candidates, which is contributed by the relevance-aware listwise reranking architecture with the well-calibrated relevance scores.

_4.4.2 Evaluation on Latency._ To evaluate the efficiency of SCaLR, we measure the average latency required to rerank the entire candidate set for a single query, comparing it to the RankZephyr, a represented listwise reranker based on LLM. To ensure fairness in comparison, we align the experimental setups of both methods and conduct the evaluations on the same GPU machine. To evaluate the efficiency of SCaLR, we measure the average latency required to rerank the full candidate set for a single query, comparing it against RankZephyr, a representative listwise reranker based on LLMs. For a fair comparison, we standardize the experimental conditions for both methods, ensuring evaluations are conducted on the same GPU machine.

As illustrated in Figure 3, SCaLR exhibits a slow linear increase in runtime as the number of candidate documents grows. In contrast, while RankZephyr also exhibits a linear growth trend, its rate of increase is substantially higher than that of SCaLR, resulting in approximately 23 times greater latency compared to SCaLR. As the number of candidate documents increases to 1000, the latency of RankZephyr exceeds five minutes, a level that is generally unsustainable for reranking systems in real-world deployments. In addition, we find the truncation length setting of RankZephyr is shorter than that of SCaLR, aligning the truncation length may result in a more significant latency disparity. We attribute this efficiency difference primarily to the autoregressive text generation approach and the redundant computations introduced by the sliding window strategy in RankZephyr. SCaLR's architectural advantage lies in requiring only a single context computation for sublist reranking, and by leveraging globally comparable relevance scores, it avoids the multiple computations per candidate imposed by the sliding window strategy.

### 4.5 Position Bias

Existing LLM-based listwise reranking methods are typically highly sensitive to the positions of candidate documents in the input [45, 48, 56]. A possible explanation for this is the "lost-in-the-middle" phenomenon, where LLMs may struggle to fully comprehend long contextual information. In our method, by introducing parallel context encoding for candidates in a decoder-only LLM,

| Orders | DBPedia | NQ | SciFact | TREC-Covid |
|--------|---------|------|---------|------------|
| Original | 50.7 | 68.1 | 76.7 | 79.4 |
| Reversed | 50.3 | 67.8 | 76.0 | 79.0 |
| Random | 50.2 | 67.9 | 75.7 | 79.1 |

**Table 4: NDCG@10 results on various input candidate orders of SCaLR on four datasets in BEIR.**

each candidate is encoded independently, without being influenced by its position relative to other candidates. This characteristic inherently addresses the position bias issue observed in LLM-based listwise rerankers.

To validate this hypothesis, we designed three sets of experiments. For each query, we reversed and randomized the order of input candidates during the evaluation phase. Table 4 presents the average results on the BEIR dataset. We observed no significant performance drops in SCaLR across the reversed, and randomized candidate orders compared with the original order setting, which is robust facing the position bias issue found in previous studies. Note that in our training data, we have maintained the original order of the retrieved candidates without changing the candidate order for data augmentation [34] to ensure robustness, which strongly demonstrates the robustness of parallel context encoding in addressing position bias.

## 5 Related Work

### 5.1 Text Reranking

Early reranking methods rely on vector space [44] or probabilistic models [42], where documents are ranked based on the degree of overlap between query and document terms. Although effective in early retrieval systems, these methods are limited in capturing deeper semantic relationships beyond surface-level term matching. With the advancements in machine learning and deep neural network, learning-to-rank [1, 3, 17, 54] and neural IR approaches [13, 28, 53] introduce new reranking paradigms to information retrieval, leading to significant improvements in ranking performance.

In recent years, pretrained language models (*e.g.,* BERT [8]) have revolutionized text understanding, significantly improving the performance of text reranking [21, 30, 32]. These models typically use cross-encoder architectures to model fine-grained query-document interactions, yielding relevance scores optimized through pointwise, pairwise, or listwise loss functions [12, 14, 62]. Various advanced strategies have been proposed to enhance the effectiveness of the cross-encoder [18, 22, 37]. A common approach involves data augmentation, for instance, leveraging query generation models to create synthetic query-document pairs for training [9, 43]. Researchers have also explored distillation strategies, where capabilities from a more powerful model are distilled into a student reranker [11, 15], as well as joint training of the retriever and the reranker [41] and long document reranking methods [10]. Additionally, some approaches utilize text generation models to estimate the likelihood of generating a query from a document, using such a likelihood to compute the relevance between the query and the document [63, 65].

### 5.2 Large Language Model based Reranking

Recent advancements in large language models (LLMs) have demonstrated their effectiveness across various natural language processing tasks [59], including the construction of effective reranking models [46, 60]. The predominant approach to LLM-based reranking employs a listwise input format, wherein the model considers multiple documents concurrently and generates a textual list of the reranked order [24, 47]. However, due to the limitations in the long context capabilities of LLMs, this method typically necessitates the implementation of a sliding window strategy during inference, maintaining the order of a sublist of candidate documents [34, 35]. Several studies have sought to enhance this reranking paradigm, such as replacing the sliding window strategy with more efficient tree inference architectures [56] or top-down partitioning strategy [33], utilizing passage embeddings for context encoding [23], and leveraging the probability distribution of the first generated token to derive sublist order [40]. In addition, cross-encoder architectures can be supplanted with decoder-only LLMs, deriving relevance scores through linear mappings [25].

Recognizing the wealth of world knowledge inherent in LLMs, numerous research has focused on achieving LLM-based reranking without extra fine-tuning, primarily by crafting specific task instructions that elicit LLMs' potential for reranking tasks. This includes employing pairwise [36], listwise [26, 46], or setwise prompting [64] strategies to utilize LLMs' instruct-following capabilities, and introducing fine-grained relevance labels to enhance the model's perception of relevance [61].

Our study adopts the listwise reranking pattern while addressing its susceptibility to local optima and inefficiencies in inference by improving both the architecture and inference strategy, ultimately achieving superior reranking performance while reducing the costs associated with listwise methods.

## 6 Conclusion

In this work, we proposed the SCaLR framework, a novel approach to LLM-based listwise reranking, addressing key limitations in current listwise reranking methods, such as local optima and inefficiencies. To implement our method, we made two technical contributions. First, we introduced a relevance-aware listwise reranking architecture, incorporating explicit relevance assessments for both list-view and point-view. Building on this, we proposed self-calibrated training to calibrate list-view relevance by point-view relevance with in-batch sampling and adaptive optimization strategies. SCaLR combines the efficiency of pointwise methods with the multi-candidate information retrieval capability of the list-view approach, ultimately achieving efficient and effective listwise reranking. Our experiments on diverse datasets highlight the effectiveness, efficiency, and robustness of SCaLR, particularly when scaling to large candidate sets. We believe that this novel listwise reranking paradigm based on large models has the potential to inspire new research directions within the community.

In future work, we will further explore the advantages brought by the proposed listwise reranking framework, such as the robustness when incorporating more candidates in listwise inputs, and investigating the impact of simultaneously utilizing point-view and list-view relevance scores.

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
