# OpenReview forum: "Self-Calibrated Listwise Reranking with Large Language Models"
_ACM.org/TheWebConf/2025/Conference — WWW 2025 Poster_

### Official Review · Reviewer_HW3v · 2024-11-29

**Novelty:** 4
**Technical Quality:** 6

**Review:**

The paper proposes a framework, Self-Calibrated Listwise Reranking (SCaLR), for leveraging large language models (LLMs) in listwise reranking tasks. Traditional listwise reranking methods face challenges due to limited context windows of LLMs and inefficiencies from sliding window strategies. SCaLR addresses these issues by introducing a process of relevance-aware listwise reranking, which incorporates explicit relevance scores for global comparisons across candidates and improving efficiency, and a self-calibrated training, which uses point-view relevance scores (assessed individually for each query-candidate pair) to calibrate list-view relevance scores, reducing biases and ensuring global comparability. Finally, SCaLR is evaluated on the BEIR benchmark and TREC Deep Learning Tracks, demonstrating better performance compared to other methods.

The paper is well-written and pretty easy to follow.

**Pros**:
1. Efficient handling of large candidate sets without compromising global ranking quality.
2. Novel self-calibration mechanism aligns point-view and list-view relevance effectively.
3. Demonstrates robustness to common reranking biases, such as positional effects.

**Cons**:
1. Marginal performance gains compared to simpler baseline models.
2. Requires extensive computational resources for fine-tuning and inference.
3. Explanation of certain methods (e.g., adaptive optimization) lacks sufficient clarity.

Minor issues:
- The definition of the set $C$ in the Preliminaries section might be slightly unclear due to the recursive phrasing, which could confuse readers. A clearer definition could be: " $C$ is a finite set of candidates, [formula here], retrieved from the corpus $P$ based on their relevance to the query $q$". One suggestion is to avoid the recursive definition of $C$ somehow, it's not intuitive.
- In formula (1) you defined $C'$ as a set of reranked candidates, that go from $1$ to the cardinality of $L$. What is $L$? You should introduce it first.
- What is the state of the art for TREC DL Tracks and BEIR? Those tables would be more impactful if you add this information
- Line 624 you wrote "superior superior performance" $\rightarrow$ "**show** superior performance"

**Questions:**

1. Your method demonstrates some innovative contributions, but there are areas where further clarity or justification could enhance its impact. While the performance improvements of SCaLR are noteworthy, they appear relatively incremental compared to existing methods. It might be beneficial to provide a direct comparison with simpler models, such as zero-shot or fine-tuned systems based on your architecture, to clearly highlight the added value of your approach.
2. The inclusion of components like adaptive optimization and point-view relevance is interesting, but their contribution to overall performance seems minimal. Exploring more scenarios where these components are removed could provide deeper insights into their necessity, similar to the discussion in Table 3, but trying to remove more than one component at a time. You could find out that some modules are closely connected with each other and not independent!
3. Finally, the complexity of the method might be perceived as disproportionate to the observed performance gains. It would strengthen your argument to justify this complexity more explicitly or demonstrate practical advantages in real-world settings. What is the computational cost?

**Reviewer Confidence:**

3: The reviewer is confident but not certain that the evaluation is correct

**Scope:**

4: The work is relevant to the Web and to the track, and is of broad interest to the community

---

### Official Review · Reviewer_wuRt · 2024-11-30

**Novelty:** 4
**Technical Quality:** 4

**Review:**

This paper presents SCaLR, a self-calibrated listwise reranking method using large language models. While the work attempts to address several important challenges, there are limitations in both methodology and experimental validation.

Strengths:
The paper presents a well-motivated solution for LLM-based listwise ranking with clear technical merits. The main contributions include a novel integration of list-view and point-view relevance scores, efficient parallel context encoding, and a creative self-calibration mechanism. The method demonstrates reasonable performance improvements over baselines through comprehensive evaluation on standard benchmarks (BEIR and TREC DL).

Weaknesses:
Despite its merits, the work has several limitations. The methodology lacks theoretical justification for key parameters (like adaptive optimization threshold τ) and sufficient analysis of computational efficiency compared to other LLM ranking methods. The presentation has notable issues including unclear architecture explanation, poorly connected mathematical formulations, and insufficient implementation details. Additionally, important technical aspects such as self-calibration stability, position bias analysis, and memory consumption trade-offs are not adequately addressed.

**Questions:**

1/How sensitive is the method to the choice of adaptive optimization threshold τ? What guidelines for selecting this parameter in practice?
2/The paper shows good scalability up to 1000 candidates. How would the method perform with even larger candidate sets? Are there any theoretical or practical limitations?
3/Could you elaborate on potential failure cases where self-calibration might not work well? Are there certain types of queries or documents where the method might underperform?
4/How does the method's performance compare with the very recent FIRST approach which also addresses efficiency in listwise reranking?
5/Could you provide more details about memory consumption tradeoffs between your method and existing approaches?

**Reviewer Confidence:**

3: The reviewer is confident but not certain that the evaluation is correct

**Scope:**

4: The work is relevant to the Web and to the track, and is of broad interest to the community

---

### Official Review · Reviewer_xPxo · 2024-11-30

**Novelty:** 5
**Technical Quality:** 5

**Review:**

This paper focuses on LLM-based listwise re-ranking. The paper proposes to use LLMs to produce global relevance scores for ranking. Specifically, it proposes explicit list-view relevance scores, enabling global comparison across all candidate documents. Also, It uses point-view relevance scores to calibrate the list-view relevance scores. Experiments demonstrate the effectiveness and efficiency of the proposed method.

Although this paper addresses some limitations in previous listwise rerankers, such as a local view for re-ranking and high computational costs, I have several concerns.

1. Vague experimental setup. Section 4.1.4 (implementation details) needs more clarification. The authors mention, "We fine-tune the model for 3 epochs." Why do the authors fine-tune the model for 3 epochs? How do they tune this hyperparameter? Did they rely on the results on the test sets to tune it?

2. Will the authors release the training and inference codes for reproducibility?

3. Limited presentation, e.g., Figure 1 is hard to follow.

**Questions:**

How does the proposed model perform compared to setwise re-rankers in terms of efficiency?

**Reviewer Confidence:**

3: The reviewer is confident but not certain that the evaluation is correct

**Scope:**

4: The work is relevant to the Web and to the track, and is of broad interest to the community

---

### Official Review · Reviewer_seJf · 2024-12-01

**Novelty:** 5
**Technical Quality:** 4

**Review:**

This paper proposes SCaLR to address inefficiencies in LLM-based listwise reranking. The idea is interesting as it combines the calibration concept with advanced LLM ranking tasks. The approach is intuitive and appears effective based on the experiments.

However, the paper would benefit from including more specific details, such as the fine-tuning process in self-calibrated training, especially since the authors have not mentioned releasing the code. Additionally, the implementation on Zephyr is somewhat limited, as the baseline includes Llama, and the methodology is claimed to be model-agnostic.

**Questions:**

1. The fine-tuning details of SCaLR in self-calibrated training are suggested to be included.
2. Regarding the challenge of point-view relevance scores, is calculating the average variance across all queries reasonable? How do you determine $\tau$? A high variance could also indicate that the query candidates are easier to discriminate, which could cause the head candidates to score higher and the tail candidates to score lower.
3. Line 461: "we exclude approximately 13% of this dataset"—what is the exclusion strategy?
4. The improvement over RankZephyr is not significant. The authors are encouraged to implement SCaLR on more backbones like Llama, as the technology is model-agnostic. Moreover, some variants in Table 3 could perform worse than RankZephyr, raising concerns about which part of the method is genuinely effective.
5. In Table 4, including a baseline result like RankZephyr would help readers better understand the extent to which SCaLR alleviates the position bias problem.

**Reviewer Confidence:**

3: The reviewer is confident but not certain that the evaluation is correct

**Scope:**

4: The work is relevant to the Web and to the track, and is of broad interest to the community